# Barriers and Facilitators Related to the Adoption of Policies to Reduce Ultra-Processed Foods Consumption: A Scoping Review

**DOI:** 10.3390/ijerph20064729

**Published:** 2023-03-08

**Authors:** Tatiane Nunes Pereira, Gisele Ane Bortolini, Roberta de Freitas Campos

**Affiliations:** 1School of Public Health, University of São Paulo, Sao Paulo 01246-904, Brazil; 2Food and Nutrition National Coordination, Ministry of Health, Brasilia 70058-900, Brazil; 3Center for Studies on Bioethics and Diplomacy in Health, Oswaldo Cruz Foundation, Brasilia 70910-900, Brazil

**Keywords:** obesity, food policy, food industry, corporate strategies, food labeling, taxes, government regulation, public policy, nutrition, commercial determinants of health

## Abstract

Cost-effective regulatory and fiscal interventions are recommended to address non-communicable diseases. While some countries are advancing regarding these actions, others have found it difficult to approve them. Aim: to conduct a scoping review to answer the question “What factors have influenced the adoption of food taxes, front-of-pack labeling and restrictions on marketing to children?”. Methods: A scoping review was developed from four databases. Studies that described and analyzed policy processes were included. Analysis was performed to identify the barriers and enablers mentioned under the guidance of Swinburn et al., Huang et al., Mialon et al., and Kingdon. Results: 168 documents were identified, describing experiences from five regions or groups and 23 countries, which have generated 1584 examples of 52 enablers (689 examples; 43.5%) and 55 barriers (895 examples; 56.5%) that may have influenced policies. The main enablers were related to the government environment and governance and to civil society strategies. Corporate political activity strategies were the main examples of barriers. Conclusions: This scoping review consolidated barriers and facilitators related to policies aimed at reducing ultra-processed foods consumption, demonstrating that factors related to the actions of governments and civil society are the main facilitators. On the other hand, as the most interested actor in promoting the consumption of these products, the strategies adopted by the companies that produce these products constitute the main barrier to these policies in all the studied countries and should be overcome.

## 1. Introduction

Risk factors related to malnutrition are the main cause of the global burden of disease and mortality, causing more deaths and disabilities than tobacco use, alcohol, and other drugs [1]. One risk factor is the consumption of ultra-processed foods (UFP), which is associated with obesity and other non-communicable diseases (NCD) [2,3,4,5,6,7]. These products are created by food transnational corporations that use corporate political activities (CPA) strategies to expand their market and protect their interests [1,8,9,10].

Obesity and CPA strategies are global health issues, extrapolating borders and demanding strong and joint action by states [11]. Cost-effective regulatory and fiscal interventions are highly recommended to address these problems, both via scientific evidence [11,12,13,14,15,16,17,18,19,20,21] and from international organizations, such as the World Health Organization (WHO) [22,23,24,25,26,27,28,29,30,31], World Bank (WB) [32,33,34], and Organization for Economic Co-operation and Development (OECD) [35].

Some countries have implemented such actions to reduce UFP consumption and, consequently, minimize impact on the economy and the population’s health. Latin America has been outstanding in the adoption of innovative solutions, improving the food environment with fiscal policies, food marketing control and front-of-package labeling (FOPL) implementation [36,37,38]. However, countries are still facing pressure from transnational corporations, as well as other sources of political inertia, which have hampered policy approval [39].

Considering that some countries have managed to advance discussions of these policies, while others have presented difficulties, the purpose of this study is to conduct a scoping review to identify and compare the barriers and enablers that influence formulation and adoption of policies to reduce UFP consumption across the world.

## 2. Materials and Methods

### 2.1. Search Strategy and Selection Criteria

A scoping review method was conducted due to the complex nature of the topic, in order to contribute to its comprehension. We followed the Preferred Reporting Items for Systematic Reviews and Meta-Analyses extension for scoping reviews protocol [40], and the project has been registered in the Center for Open Science database [41].

A search strategy to answer the question “What factors have influenced the adoption of food taxes, FOPL and restrictions on marketing to children?” was developed, and the literature search was undertaken in Medline (PubMed), Scopus (Elsevier), Web of Science (Clarivate Analytics) and Virtual Health Library (Biblioteca Virtual de Saúde—BVS, in Portuguese) databases in April 2022, using the search terms (MeSh for the English search and DeCS for Latin American and Caribbean Center on Health Sciences Information) presented in Table 1, with no restrictions. To complete the search and prevent important papers from being excluded, the references from the selected studies were also revised.

A screening of the retrieved literature was carried out by reviewing the titles and abstracts that met the inclusion criteria:

(a) Descriptions and analysis of the formulation processes of the three policies.

(b) Analysis of barriers to and enablers of approval for the policies.

(c) Analysis of strategies used by stakeholders to approve the policies or delay and disrupt approval.

(d) Description of support for the policies by stakeholders and the population, included only if we found some other document that had described the related political process in the same country.

In addition, documents concerning processes related to NCD or obesity or nutrition policy have only been included if the regulation norm was unified for more than one policy or if another document describing at least one of three policy processes from the same country was included. In the cases of intergovernmental political spaces (Southern Common Market—Mercosur, European Union—EU, Codex Alimentarius Commission, and World Trade Organization—WTO), we included only Member States that had already been included individually in other documents.

The following kinds of study were excluded: (a) theoretical and ethical discussions of policies; (b) policy evaluation (implementation or impact); (c) consumer understanding and opinion on specific policy models (e.g., comparison of traffic light and warning labeling); (d) policy characterization (e.g., extension and type of marketing practices); (e) other policies, such as on tobacco and alcohol, without comparison with one of the three studied food policies. In addition, we excluded studies that addressed restriction of marketing to children under two years, marketing restrictions in institutional environments or in specific contexts (e.g., schools, markets, sport events), and menu labels or specific food and beverage labeling (e.g., FOPL in soda).

All studies were included in the Mendeley^®^ software, where duplicates were identified and excluded. In the results section, we use a code to refer to the documents we found in this review, in which the letter represents the policy described in the document: T = food taxes; L = food labeling; M = marketing restrictions, and, in parentheses, the number identifies the reference found in the review (e.g., “T(1)” means the first document regarding food taxes). For documents describing more than one policy, more than one letter was used in the code (e.g., “LM(1)” means the first document that describes FOPL and marketing restrictions). The selected documents and their respective codes are referenced in Appendix A.

### 2.2. Analysis

The analysis was developed as follows:Consolidation of the main documents and studies’ characteristics: identification of papers’ characteristics (authors, year of publication, and type of document); study subject; studied policy, country and period covered by the study; main scope (experience or support paper) and a summary of the results identified as important for the policy formulation and its approval. For definition of period covered, date of document analysis was considered or, if not mentioned, the first and the last year cited or interviewed were identified. When the documents addressed more than one policy or NCD, obesity or nutrition policy, the results were replicated for all policies described, except when the authors detailed facts related to a specific policy. The data were summarized in an Excel^®^ spreadsheet.Descriptive analysis of the main documents’ characteristics: the descriptive analysis included the distribution of relative and absolute frequencies.Classification of the data into pre-established categories: deductive coding of data, where the information of the summary related to the policy process described in Step 1, had quotes reproduced and was classified in pre-established categories, according to the contextual elements or stakeholder practice, strategy, or discourse. The categories were adapted from: International Network for Food and Obesity/NCDs Research, Monitoring, and Action Support domains of infrastructure component [42]; levers to overcome policy inertia and address The Global Syndemic [11]; demand-side strategies to mobilize obesity prevention policy described by Huang et al. [43]; framework for classifying the food industries’ CPA adapted by Mialon et al. [44]. Regarding the political and economic environment, we identified elements described by Kingdon [45] that had not been included in the other frameworks.New categories and adaptation of predefined categories to best reproduce the review findings: Deductive coding was also used to allow new categories to emerge, leading to the reorganization, cluster, and creation of categories. Discursive strategies were separated from the instrumental strategies, as Mialon et al. [44] proposed. However, they were classified in a group of discourses used by opponents (and not only the food industry), based on Mialon et al. [44], and those used by advocates to promote the policy were identified, constructed by the authors, following a similar structure. Agreement about the categorization of examples was reached after review of authors. The study’s final coding frameworks, with detailed descriptions of each code (domains and subdomains), are provided in Figure 1, Appendix A. These categories are not exclusive, as some examples could be classified in more than one group.Descriptive analysis of enablers and barriers: After definition of the categories, each quote was reread to classify whether, in the country analyzed, the category was an enabler (Appendix A), a barrier (Appendix A) or a discursive strategy (Figure 1). From these data, which had already been stratified by country and policy, a descriptive analysis was performed, including the calculation of the distribution of absolute and relative frequencies. The absolute frequencies were calculated by summing the country examples (units) of barriers or enablers by policy and domain. The relative frequencies are represented by the percentage of each domain over the total barriers and/or facilitators. Thus, a sum was performed, and the percentage of cases was calculated by identified factors (barriers and facilitators), according to the domain of the framework.

## 3. Results

In total, 168 documents were used for data extraction and citation, comprising 76 documents for food tax, 26 for FOPL, 17 for marketing restrictions, and 49 for more than one policy. Figure 2 demonstrates the numbers of records returned from the literature searches for each policy and the process of selection that resulted in the 87 articles selected, added to 32 documents included after analyses of the references in the selected literature. The other 49 studies are related to documents that approached more than one policy (25 from the literature review and 24 from the references review).

Although the search does not have limits, all selected documents were published after 2005, and the majority (78.6%) were published after 2013. Countries’ experiences represented 77.4% of the documents, mainly with an approach of up to five years (66.7%).

Aligned with the search strategy, more than two thirds (65.5%) have as subject description the policy process, its barriers and facilitators, the lessons learnt, and strategies used by the main stakeholders. Experiences from five regions or groups of countries (Commonwealth, European Union, Latin America, and United Kingdom) and 23 countries were identified; USA (17.6%), Australia (14.5%) and Mexico (9.0%) were the most studied (Table 2).

We have recognized 18 country experiences related to food taxes, 16 for FOPL, and 12 regarding food marketing restrictions (Table 2). These experiences have generated 1584 examples of 106 factors that may have influenced the discussion of the three policies, which were classified into 52 enablers (689 examples of positive factors by country; 43.5%) and 55 barriers (895 examples of negative factors by country; 56.5%)—Appendix A (Table 3).

For all policies, the main enablers seem to be those related to the government environment and governance, representing almost half of the positive examples (49.8%), and to civil society strategies (30.2%). On the other hand, CPA strategies used by the private sector, chiefly the food industry, are the main examples of barriers for all policies (63.7%), ranging from 61.3% in food tax cases to 67.1% in FOPL cases (Table 3).

In the next section, the results will be detailed by subclassification. with a description of some of the examples found.

### 3.1. Contextual Factors

Contextual factors related to the country’s political and economic environment (e.g., conservative, progressive or stakeholder’s ideological positions; elections that resulted in changes of government; population or media support) and knowledge and awareness (evidence produced and disseminated and amplification of discussion by the media) were identified as having been important to the policy process, regardless of whether as an enabler (14.5%) or a barrier (9.1%) (Table 3).

A clear example of how one of these factors—the political-economic environment—could affect policy approval and how this could be different for each policy is illustrated by Mexico, which has approved the three policies. The approval of a sugary drinks tax occurred in a conservative government, which had tax reform in its policy agenda. The food taxes were a civil society initiative promoted by a conservative parliamentary leader that had support from other conservative parties^T(6,15,18,38)^, and opposition from progressive actors^L(25)^. On the other hand, the discussion on approval for a law to adopt FOPL and to restrict food advertising was suppressed by conservative parties, and its progress only occurred after presidential elections that switched both the president and the majority in the Mexican Parliament to progressive parties^L(25)^.

The conservative scenario also appeared as a barrier for FOPL and food marketing regulation in Australia^M(16),L(15),LTM(1,37),T(63)^, Chile^LTM(8,23)^, Colombia^LTM(7)^, France^L(12)^, Fiji^M(10,14)^, South Africa^LTM(11)^, and Uruguay^L(1)^. In addition, in Australia^M(16,17),LTM(10)^, Chile^LTM(8)^, Colombia^LTM(7)^, and Uruguay^L(2)^, a progressive environment and decision makers’ positions contributed to FOPL and marketing restrictions discussions or approval. Conservative governments were mentioned as being only positive for food taxes (Denmark^T(5,73)^, France^T(42)^, UK^T(12,31)^), but not in all cases. In the United States, for example, a comparative study of local policies demonstrated that the cities making major efforts to approve the policy, independent of its success, were dominated by the Democratic Party^T(55)^. The same relation was seen in local news: left-leaning local newspapers were more likely to cover the taxes policy debate in the USA^T(49)^.

Regarding media support, among the cases where the media position was mentioned, most of the news was negative, especially about food taxes. The news highlighted the possible impact on national and international trade^LTM(8),T(16)^; framed it as a policy that would be against individual freedom^LTM(8),T(51)^; questioned the public health rationale^T(42)^; and presented more food industry and citizens’ unfavorable comments^T(5,16,42)^. This discourse is aligned to the private sector, as we will demonstrate later, and, in some countries, such as Colombia and Fiji, the UFP and beverage industry not only influenced, but also owned some media organizations^LTM(7,29),T(26)^.

Population support and its construction can also be critical for the policies’ approval. In that regard, the studies demonstrated that it is especially important in countries or provinces that considered public opinion in final decisions, as in some USA cities where ballots were conducted with the citizens and defined the policy’s destiny^T(39,46,55,74)^. In countries where social participation is promoted by other mechanisms, such as public consultations, population support could be important in putting pressure on the government, based on public contributions. Thereby, data demonstrating support is also necessary. Many studies focusing on evaluating public support for the policies were found in this review^L(6),LT(1),LTM(3,5,15,20,30−31,33−34,36,38−39,41),M(1,6,15),T(1,4,7−8,13,17,19,23,28−29,45,50−51,58−59,61−62,64),TM(1−3)^, and there was more support (2.6%) than opposition examples (0.7%), the latter concentrated on food taxes (Appendix A).

Population support can be related to specific characteristics, such as gender, level of education, political-ideological position and knowledge of health and nutrition. Women were more supportive of the taxes^T(17−18),TM(2)^, FOPL^LTM(3,31)^ and marketing restrictions^TM(1−2),T(28),LTM(3,31,36)^ than men. Those with higher education were more supportive of taxes^T(18,28,59)^, FOPL^LTM(3)^ and marketing restrictions^LTM(36)^ than those with lower education. Furthermore, a lower socioeconomic status was associated with support for food taxes in Australia^LTM(33)^, New Zealand^T(67)^, and USA^TM(1−2)^. The same was found for marketing restriction support in Australia^LTM(36)^ and USA^TM(1)^. In addition, progressive ideals and a defense of the regulatory role of the state, distrust of industry and the recognition of its role in obesity were revealed to influence support for food taxes ^T(4,17,19,28−29,39,50,62,69),TM(3)^, and to marketing restrictions^M(6,15),T(28),TM(1,3)^; and awareness of UFP consumption, obesity, health, and policy impact seems also to affect food taxes ^LTM(31,32),LT(1),T(15−18,39,45,47,53,58,60,64,69,75),TM(1)^, the FOPL^L(6),LTM(31,32)^, and marketing restrictions support^LTM(31,32),M(6,17),T(47),TM(1)^ in some countries.

Another element that seems to be fundamental to the policies’ approval is the existence of evidence related to the topic, that is, nutritional status and food intake monitoring, obesity and UFP consumption consequences, policy evaluation (implementation and impact) in other countries, and projection of local policy impact and its cost-effectiveness. In almost all the cases, the evidence availability appeared as a facilitator to the discussion of and to approval for food taxes (66.6% of cases), FOPL (75.0% of cases), and marketing restrictions (83.3% of cases) (Appendix A).

Presenting the consequences of obesity and UPF consumption to decision makers and to the public is important and can motivate the discussion, as occurred in Ecuador^LT(3)^, Mexico^LT(3),T(60)^, and Canada^LM(2)^, but is not enough. Decision makers seem to consider economic arguments as more crucial, such as the burden of obesity and NCD on the country, and the cost-effectiveness of policies, which regards savings for government, society, and the possible impact on industry (e.g., job losses and sales impact). In some countries, this specific evidence was absent, as in Australia and USA, where a cost-effectiveness analysis was required for actors potentially affected by the proposed regulatory policy^M(3),LTM(10,37)^. However, in countries where evidence was available and used by the policies’ promoters^LTM(1,17)^ or produced during policy discussion, as in Chile^LTM(14)^ or Mexico^T(18)^, the debate was facilitated, providing the basis for counter-arguing against industry criticism.

### 3.2. Government Environment, Strategies, and Practices

Elements most described as facilitators in policy processes were related to government environment, strategies, and practices (52.2% of FOPL; 50.3% of food taxes; 46.5% of marketing restrictions examples) and were also the second most described as a barrier (25.6% of marketing restrictions; 25.2% of food taxes; 22.4% of FOPL examples) (Table 3).

Comparing the policies, the factors related to the government environment (policy agenda, prioritization of health in all policies, parliamentary, technical team, and government position) were identified as more influential on food taxes cases (24.5% of enablers and 10.5% of barriers), although repeatedly described in all policies (Table 3).

The recognition of the problem (obesity, malnutrition, unhealthy eating habits) and the entry of this problem and its proposed solutions (marketing restrictions, food taxes and FOPL) into the political agenda, as well as the definition of goals and actions related to these issues, were elements described in almost all policy processes. In some countries, such as Australia^L(11,20),LTM(10,40)^, and France^L(9,10)^, obesity or healthy eating were recognized as national health priorities, resulting in several reports, the creation of commissions, and formulation of national plans that expressed the political will to face the problem, building a favorable environment for FOPL approval, even though voluntary. In other cases, such as Australia^T(63)^ and Denmark^T(5)^, taxes expanded the public debate, as they became the subject of electoral dispute. A budget crisis in the countries or the discussion of the need for fiscal reforms were also facilitators present in half of food taxes’ cases (Appendix A).

In addition to being part of the political agenda, an explicit position of the Head of State or of an important minister or secretary could also promote or delay policy approval. For a tax’s approval, the need was identified for the economy leader or the Head of State support, regardless of the health minister’ position. An example is Mexico, where the SSB tax was imposed by the Minister of Finance, despite the support of the Minister of Health^T(15,18,38)^, who was a member of an institution funded by a transnational food corporation and promoted discourse aligned to the food industry^T(15)^.

However, high-level leaders of the economy, trade, agriculture, and commerce were recurrently mentioned as major opponents inside governments. They used arguments such as questioning policies’ efficacy and their economic impact, and defended economic powers, even lobbying for them^LTM(1,14,21,23,29),M(4),T(32,33,42)^, not only in the specific country, but also in international arenas, for example, discussions about FOPL in the Codex^L(11)^.

In this context, a change of government position, induced by any other factors, could define the policy’s future. This situation can be exemplified by the cases of Denmark and UK taxation. The Danish government approved a fat tax, and this was strongly defended by the Minister of Taxation during discussions, who framed it as a public health policy that could contribute to better food choices^T(73)^. Nonetheless, during the electoral process, due to political pressures, the same party and the minister switched their position, questioning the policy, asserting that the new liberal priorities were irreconcilable with a fat tax and declaring that the taxation was a mistake^T(5,33,73)^. In the UK, the change was positive for the sugar sweetness beverage (SSB) tax. The government has altered its position, initially aligned with the food industry and contrary to the policy, but then defending it, after new evidence was published and after civil pressure, especially from celebrities, implying approval for the taxation^T(12,36)^.

Additionally, the parliamentary position is also important, mainly for policies and countries that require discussion in Congress as a prerogative of approval and implementation. This was the case in Colombia, which experienced conditions for approval of SSB taxes, but this was defeated in Congress, influenced by the food industry lobby and funding^LTM(7)^.

Governance, which includes elements of leadership, the existence of mechanisms of transparency and use of evidence, accountability, and legal support for policy implementation, was mentioned in the documents as influencing all the policies, representing more than a quarter of the enabler examples and more than 10.0% of the barriers (Table 3).

Leadership in conducting and promoting the policy was one of the main factors identified as a facilitator in the government category. High-level leaders who acted to promote the policies and had a significant role in discussion were mentioned in the government of Australia (Health Minister)^LTM(1),T(63)^, Chile (Health Minister)^L(16),LT(3)),LTM(14)^, Colombia (Minister of Health and Social Protection)^LTM(7),T(26)^, France (Minister of Budget)^T(42)^, and South Africa (Health Minister)^T(16)^.

Moreover, cooperation between leaders of the executive and legislative branches was cited as crucial to policy approval in Chile^LT(3),L(16),LTM(14)^. Thereby, parliamentarians act as leaders, promoting and fighting for the policies. In Mexico, the SSB tax was formulated in the Senate^T(53)^ and was only approved after an agreement between the main Mexican parties to include it in the fiscal reforms^T(18,38)^. In the Philippines, a lawmaker proposed a SSB tax. The first attempt was unsuccessful; nonetheless, she resubmitted, ensuring the health and finance decision-makers’ support, and the proposal was approved^T(52)^.

Transparency comprises elements related to the possibility of society participating in policy development, from open discussions with society (e.g., public consultations) to mechanisms to prevent conflicts of interest (COI) in the policy process. Lack of transparency, especially the absence of procedures restricting commercial influences on decision makers and the promotion of partnerships with the food industry, including in working groups related to policy design, corresponded to the main government barrier identified in this study (70 examples of barriers). In all cases, except Canada, Iran, Ireland, and Saudi Arabia, at least one of these barriers was mentioned (Appendix A).

The promotion of open discussions of the policy between society and government (e.g., seminars, consultations, public hearings) was positive in the studied policies. These discussions could amplify discussion, and allow any member of the society, which includes the food industry, to make its contribution and suggestions^M(5,7),L(8,16),LT(3),LTM(4,6−9,14,21,35),T(12,16,18,39,41,46,51,53,55,74)^. We highlight the USA’s cities and states, where the local population was responsible for the final decision through ballots, increasing debate and a role for the population in the process, even when the decision was negative^T(39),T(46,51,55,74)^.

However, the government allowed the use of food industry strategies to influence the three policies outside of the common spaces of civil society, for example, receiving funds to policymakers and political party campaigns^LTM(7,17,21,27),M(3),T(55,63,70)^, opening spaces to create relations and receive lobbyists^L(10,17,24),LT(2,3),LTM(1,7,8,22,23,25,27,29,35,37),M(3,12),T(5,15,16,26,34,40,43,52,55,64,67),TM(26)^; offering positions in the government or allowing members of government to defend food industry interests^L(17),LT(2),LTM(17,21,23),M(3),T(15,38,40,54)^. Furthermore, the opening to industry participation was promoted by government through partnerships, regulation substitutions and voluntary actions^L(11,14,17,19,20),LTM(1,18,20,24,28,34,37)^, and the food industry was included in strategic workgroups without establishing mechanisms to prevent COoI^L(25),LTM(40,26,28),M(3),T(5,16,33)^.

On the other hand, to avoid the food industry presence in the formulation steps of policy and to define mechanisms to prevent COI were seen as facilitators of policy approval. In Canada, communications relating to the Healthy Eating Strategy between the Health Ministry and other stakeholders were published on the government website^LM(2)^. In Chile, a Law on Lobbying and a Law on Transparency, which oblige that any meeting must be requested through legal processes and has to be publicized, and the fact that the food industry did not participate in the technical committees responsible for making decisions on FOPL criteria, were seen as elements that contributed to approval^LTM(9,23)^.

In addition, the creation of structures and mechanisms for interactions with or support from civil society on policy design was identified as positive for all the studied policies. This is a category of coalition management interrelated with transparency and the use of evidence. In general, these spaces comprise stakeholders who have personal (e.g., consumers) and technical experience, along with independent experts^L(10−11,19,25),LT(3),LTM(8,16,35,40),M(14),T(5,15,26,43,53)^. In Chile, a workgroup developed the scientific basis and provided recommendations for defining all technical criteria of the FOPL and marketing regulations^LT(3),LTM(8)^, for example. 

Another interesting factor that appeared as a facilitator related to this topic was a surprise element in the final decision. During discussions on taxation in France^T(42)^, Mexico^T(38)^ and UK^T(12,36)^, and the deliberation on FOPL and marketing restrictions in Chile^LTM(8)^, not communicating the decision to stakeholders before official release was positive for the policy outcome, because it confused, and created an extra difficulty for, opponents’ action.

Regarding coalition management, which was mentioned in less than 7% of enabler examples and less than 4% of barrier examples (Table 3), after the creation of structures and mechanisms for interaction with civil society, the second most cited element was cohesion in the government, both as a barrier and as an enabler (Appendix A). In Chile, the consensus between involved sectors was crucial for policy approval, because it created a solid foundation for FOPL and marketing restrictions^LTM(14)^. However, in Australia, the complexity of the food regulatory system was considered a barrier to approving food policies^LTM(10,37)^.

Opponents inside the government can hinder the efforts to approve policies, as described by government positions. Moreover, in some cases, members of government, including parliamentarians, also implemented well-structured strategies to suppress the measures, for example, not recognizing their responsibility for regulation, transferring this to other actors^M(2)^, negotiating agreements to defeat the policy^T(26)^, altering process flows to include a new commission or analysis that would defeat the policy^T(44),LTM(7)^, and cutting budgets to regulaton agencies^M(3)^.

Other factors related to the government were the existence^LTM(10),T(37)^ or absence^L(13,24),LT(3),LTM(8,37,40),M(3),T(44,64,73)^ of bureaucratic resources, capacity (number and skills) and a budget that could decrease or increase the possibility of approval of the policies, respectively, and difficulties in defining policy criteria (e.g., no clear products for regulators to target^LTM(1)^, defining unhealthy foods^LT(3)^ and relevant tax scope^T(42)^, factors related to monitoring and implementation^LTM(14),T(4)^, but these were less commonly mentioned in the documents.

### 3.3. International Organizations’ Role

Regarding international organizations (e.g., Pan-American Health Organization—PAHO, WHO, Food and Agriculture Organization of the United Nations—FAO, the WB), their support was mentioned as important for policy approval or discussion in 4.8% of the examples, varying from 4.2% of food taxation examples to 5.9% of FOPL examples (Table 3).

The main action identified was visible support for the policy before its approval, whether in speeches by organizational leaders directed to the country or in technical reports. Compromises led by the WHO and ratified by the members and technical meetings or consultations induced discussion of obesity and its solution in Australia^LTM(1)^ and Chile^L(16),LTM(8)^, and resulted in the first proposal of the law that was approved in this latter country^L(16),LTM(8)^. In Colombia, the Director of the PAHO for the Americas sent a letter to parliamentarians supporting the taxation project that was being discussed^T(26)^, and in Ecuador the Director participated in a high-level meeting, increasing media attention to the FOPL discussion^LT(3)^.

Another way that international organizations supported countries was by being part of groups related to the policy and providing technical support and advice to policymakers, based on scientific evidence. In Chile, FAO, and PAHO directly contributed to the FOPL and Marketing Act and its Regulations^LTM(14)^. In Colombia, the WB made a significant contribution to the tax proposition^T(26)^. In Mexico, PAHO had active participation, not only providing technical support^T(15,38,53,75)^, but also building alliances with various stakeholders in taxation cases^T(38,53)^.

### 3.4. Civil Society Strategies

The context and the strategies and practices used by civil society were the second most common facilitator of policy discussions and approval, representing 30.2% of the positive examples by country (Table 3). The strategies related to coalition management, i.e., creating relationships with the media, opposition to, fragmentation and destabilization of opponents, constituency fabrication, seeking involvement of the community, leadership, and funding, seem to have been the most important in this group (15.0% of marketing restriction, 13.3% of taxation, and 10.3% of FOPL examples).

In all policies, monitoring and exposing strategies of the CPA realized by the food industry and other opponents were recurrently mentioned in the documents (Appendix A). This facilitator occurred in different ways: conducting research about policy impact in industry (jobs, for example) or about the proposal’s efficacy (non-regulatory and voluntary actions, FOPL models) and using the data to counter-argue against opposition discourse^L(10,25),LTM(6,14,19,20,35,41),M(5,12,16),T(3,22,36,75)^, creating platforms where illegal or unethical CPA complaints could be made by citizens^LTM(35,40)^; and publicly exposing the inherent COI of food industry participation in policy decision^LTM(7,35),M(7),T(15,29,49,74)^, the links between government members or entities and the food industry^T(15)^, and the other strategies used by the food industry, such as its legal actions against the State, food market dominance and economic power^L(25),M(10),T(29,51)^.

The creation of broad-based and well-coordinated groups was a constant facilitator mentioned for all policies. Some common characteristics of their organization were a diversity of members with scientific, legal, political and marketing experience, the definition of consensus about principles, policy priorities and strategies according to the political context, with clear division of duties and discourse alignment^L(20,25),LT(3),LTM(7,35,40),M(17),T(15,16,18,38,53−55,60,66,75)^.

In successful cases, an accurate identification of window opportunities (e.g., elections, crises, reforms) and a consequent and timely development of strategies, relationships, and mobilization of society through simple message campaigns seem to have been fundamental to policy results^L(25),T(18,53−54,75)^.

In Mexico, the civil society group, Alianza por la Salud Alimentaria, analyzed the political context and was able to identify tax reform as a timely window of opportunity and agreed to focus on the tax on SSB as a unique policy priority, framing it as an important economic and health measure^T(6,38,75)^. They also mapped potential supporters before the election and recognized a political leader from the conservative opposition to introduce the proposal and to advocate for the tax. In parallel, they identified that the health secretary had COI and that it would be more strategic to obtain support from the finance secretary, who was in favor of tax reform. Thereby, they constructed a powerful coalition behind the measure^T(18,38)^. This is also an example of strategically obtaining indirect access to decision makers through advocacy and having supporters in strategic positions. In addition, the Alianza paid for mass media campaigns before and during the evolution of the SSB tax debate. All messages were tactically studied and covered the nature of problem, its causes and solutions, anticipating arguments from opponents and their strategies^T(18)^.

To make these actions possible, establishing partnerships to increase public capacity and financial and regulatory resources, along with maintaining transparency in their funding sources, was crucial. In Mexico, they had support from Bloomberg Philanthropy, allowing them to contract a lobby group and develop a national and effective campaign^T(18,38,54),L(25)^. This institution also financed similar actions in Berkeley and Oakland (California/USA), Boulder (Colorado—USA), and Philadelphia (Pennsylvania—USA)^T(55)^. External financial support from other institutions was further mentioned^L(25),T(16,55)^ from their own country, as happened in Australia, which funds VicHealth to support research, programs and advocacy for public health^LTM(40)^.

The food industry invests heavily to oppose the measures, as we will demonstrate in the next section. Therefore, the existence of funds to counter their efforts seems to be necessary for civil action, and the lack of resources has seemed to be a barrier in many countries. In Australia, during the FOLP discussion, these limitations implied a lower participation during technical and political negotiations^L(20)^. In Fiji, there were difficulties in sustaining advocacy and maintaining a marketing campaign over time^M(14)^. Other documents cited that the difference between the resources of the industry and those of civil society significantly hampered tax defense in the USA^T(39,43,55)^.

Lack of cohesion between members of civil society was also cited as a barrier. This was described as insufficient consensus on the causes of and solutions to obesity^LTM(1),T(63,73)^ or on policy definitions^L(23),T(42)^, divergent messages communicated by public health advocates, ambiguous positions^M(17),T(36,42,44,55)^, and different opinions on receiving food industry funding or participation in civil events^L(14),LTM(1,34),M(10),T(38)^.

Other strategies identified were to establish close relationships with media organizations, journalists, celebrities, and bloggers; and to nominate or promote leaders from civil society who were able to mobilize society and promote the opening of windows of opportunity. The former was mentioned more often in the case of taxes. The partnership with Jamie Oliver to promote the sugar tax in the UK, for example, seemed to be critical to the discussion. This is also a case of having a leader in civil society who used their worldwide public recognition to promote a food policy^T(9,12,63,66)^. However, champions were not only celebrities, but also and mainly advocates from academia and NGOs. In Mexico, two leaders from civil society (consumers and academia) have had an important role in the tax approval^T(15,18,53−54,75)^, for example.

A few less often mentioned strategies relating to coalition management were to procure support of community and business groups (e.g., producer organizations^LTM(35),T(18)^ or retail stakeholders^T(35)^ less impacted by the measure); and to involve citizens in political campaigns (e.g., through on-line platforms^LTM(35,40),M(17),T(16)^) or digital petitions and advocacy^L(7),LTM(35),T(67)^, social media campaigns^LTM(7),T(16,18,53)^, and public acts^T(18,53)^.

The direct involvement of and influence on policy by civil society was the second most identified strategy that contributed to FOPL (9.4% of examples) and marketing restrictions (10.0% of examples), regarding discussion and approval of different cases (Table 3). For these two policies, participation in policy development including advocates in the government decision-making process was the most usually mentioned facilitator (Appendix A). This was made mainly through official invitations to join expert groups or participate in official events and meetings, providing assistance to policymakers and contributing to public consultations based on evidence^L(8,10−11,14,24−25),LM(2),LT(3),LTM(1−2,7−8,14,19,35,40),M(3,5,7),T(5,15,18,26,38,43,53,55,73)^.

Information management was also important for all policies, especially in funding research, and developing and disseminating proposals and arguments based on scientific evidence without COI and supported by academics and health professionals, representing about 9.0% of examples (Table 3). Much of the research relating to the policies cited in the documents found and described earlier were generated in civil society through academics, who promoted investigations, elaborated reports, and participated in and hosted scientific events to disseminate findings.

This facilitator category includes social marketing campaigns. Campaigns were more cited in cases of taxation and marketing restriction. However, the messages were similar. Civil society disseminated information about healthy eating^L(24),LTM(7),T(16,18,27)^; the impact of sugar intake or harmful effects of SSB on health^LTM(7),T(18,26,53,63)^; and direct policy defense^L(25),LTM(7),T(18,27,57,38,75)^ via TV, billboards, radio, full-page journal articles or/and social media channels.

Another less cited strategy was to simplify, refine, and adapt messages to decision-makers. In Australia, the Obesity Coalition created summaries for regulatory priorities^LTM(40)^, and, in Mexico, the National Institute of Public Health assisted with scientific findings’ translation for policy makers^LT(3),T(75)^, for example.

### 3.5. CPA Strategies

The strategies used by the private sector, especially food industry corporations, were the most mentioned barriers for all policies and countries, corresponding to 67.1% of FOPL, 63.2% of marketing restrictions, and 61.3% of tax examples. Barriers relating to coalition management were the most identified in the documents, followed by information management and involvement in and influence on policies. Legal actions were described less, despite being relevant (Table 3).

The private sector promoted public–private interactions with health organizations. Some examples of these interactions were the Food and Health Dialogue, a partnership with the government to promote food reformulation on a voluntary basis; the Healthy Food Partnership, which influenced FOPL development and implementation in Australia^LTM(1,14)^; and health campaigns and the supply of drinking water in schools by beverage and other food industries in partnership with the Mexican government during and after discussion on taxes and FOPL^LT(2),T(15,38)^.

Access to policy elites and other key opinion leaders was also a common strategy mentioned in the documents. A close relationship between the food industry, parliamentarians and officials close to ministers was mentioned as a barrier to taxes and to FOPL discussion in Australia^LTM(27,28)^. In Brazil, Colombia, and Mexico, parliamentarians were approached by the food industry at critical moments of the policy discussions, and presented its defense and proposals to the plenary^L(17),LM(35),LT(2),LTM(21),T(15)^. Industry has also realized conferences and created workgroups about the policy proposals, which had the participation of ministers ^LTM(17,29,35),T(5)^, and promoted visitations of ministers, presidents or influential government employees to their factories^T(15,34,67)^.

In addition, relationships with nutrition and public health associations were identified, highlighting the food industry’s role in FOPL development ^L(14)^, inviting them to technical meetings^LTM(21)^, and participating in workgroups created by the food industry^L(18)^. The food industry has supported professional organizations, including funding or advertising in their publication^L(25),LT(2).LTM(23,24,26,28),T(15,38)^. The documents mentioned that some of these organizations publicly took a stand against the policies without making explicit their relationship with the industry, as described in Mexico, where the Mexican Federation of Diabetes positioned itself against the soft drink tax, without revealing the funding received from Coca-Cola^T(15)^.

Besides policymakers, health professionals and other key opinion leaders, the private sector sought support from the community by undertaking corporate philanthropy, funding events, community-level, environmental and physical activity initiatives in almost all cases (Appendix A). Another strategy described in the documents was the establishment of fake community grassroots organizations. The most commonly mentioned and well-known case is that of the group Americans Against Food Taxes, created by the food industry to prevent tax approval across the United States^T(48,76)^. However, other local groups were identified ^LT(2),T(16,40,51,53,55)^. Research organizations created and sponsored by the food industry were also mentioned^T(21)^.

As well as civil society, the other group that the food industry has established relationships with is the media. However, unlike policy advocates, the food industry is powerful, pays for advertising and, in some cases, owns media channels. A considerable barrier in the Colombia discussion of SSB taxes was the access of the food industry to media organizations. Some of the main media groups belong to an enterprise which owns a SSB company^LTM(7),L(17)^, blocking public discussion in the media^T(26)^.

A significant strategy of the food industry is cohesion inside the group. Unlike cohesion in government and civil society, which was classified as an enabler, food industry group alignment was identified as a barrier, and their lack of cohesion and coordination a facilitator to policy approval and discussion. Misalignment between political opponents in France^T(42)^ and the UK^T(12)^ had a positive impact on tax approval (Appendix A). However, the most common situation was the opposite. In almost all cases, we identified that the food industry had obtained private support, from international groups to food manufacturing producers and small businesses, and from governmental groups, to oppose the measures, creating cohesion between interested actors against the policy and aligning its discourses with other cases (Appendix A).

In addition to these strategies, the food industry has likewise used strategies of opposition, fragmentation, and destabilization, especially by discrediting or threatening public health advocates. The documents reported questioning of ministries and regulatory agencies’ intentions, mandate, role, and transparency mechanisms—even when the food industry had participated—and criticisms of the credibility of the policy process^L(16,19,25),M(3),LM(2),LTM(4,26,35),LT(2),T(4,27,42,46,51,73)^. Discrediting and intimidation of public health advocates and agencies included direct threats and tactics to affect careers and research projects^L(8,25),LTM(25,27,28),T(72,73),T(15,34)^. In countries where civil society received support from international institutions, these were also publicly criticized, claiming that they represented business interests^L(19,17),LTM(25),T(51)^.

Moreover, some documents mentioned that the food industry had created antagonism between professionals^LTM(1,22,27,35),T(53)^. Although it has been cited less often, all the strategies and tactics for promoting relationships with health professionals and the production and amplification of information, described below, can reduce agreement between health professionals and create confusion regarding the science.

These strategies of criticizing evidence and suppressing the dissemination of research or data that does not fit the industry’s interests are related to information management. The food industry has questioned evidence used by policymakers and has asked for new and local evidence about the relevant policy ^L(8,19),LTM(1,21,22,24,35),M(11),T(41)^. They have also confronted the impact of the proposed policy on health^M(11),T(5,49,53,72),LTM(4,6,21,24),LM(2),T(41)^, the role of saturated fat, sugar or UPF and its ingredients obesity and NCD^L(2,16,19),LTM(4,23,28,35),T(41,52,63,72,75,T(44,60)^, and the negative impact of advertising on children’s health or food choices^M(11)^. In Chile, there were examples of how the food industry suppressed results and withdrew funding from institutions that had published negative data against them^LTM(23)^.

In addition to these strategies concerning information management, which represented about 20.0% of the example of barriers for all policies, the most mentioned tactic was cherry-picking data that favors the industry, including use of non-peer reviewed or unpublished evidence (Table 3). The food industry has used data with no scientific evidence base or data from non-academic, non-peer-reviewed reports, often funded by themselves, to defend their policy preferences, discredit arguments from public health advocates and criticize policy proposals^LTM(4,17,19,21,23,26−29)M(3,12),T(3,5,15,21,24,25,36,41,72,73,75), L(2,17,19),LM(2),LT(2)^.

For example, in Canada, during the FOPL discussion, a food group association presented a report containing data about public comprehension of the label, without requiring the approval of a new model. However, a study identified that 35% of the citations used to support the food industry argument in a public consultation about national food policy were not peer-reviewed and 43% received funds from the food industry^LM(2)^.

Several citations were found relating to the funding of research, including from academics, ghostwriters, their own research institutions, and front groups. For example, in Australia and the USA, research groups have been funded by the food industry to focus on the importance of physical exercise, reducing the role of diet in obesity prevention^T(21,63)^. Another well-studied research group funded by the food industry is the International Life Sciences Institute (ILSI), which was founded by Coca-Cola, receives funding from various food companies, and has representatives in many countries, to realize and publish studies favorable to their financiers^LTM(23)^. To disseminate the information produced or funded by the food industry, it promotes and participates in scientific events, besides proposing education, especially to students, health professionals, teachers, and schools. In Colombia, for example, the ILSI promoted one event on FOPL targeting students in nutrition, and another, in partnership with the Colombian Association of Dietetics and Nutrition, aiming to shape public opinion ^LTM(21)^.

Another strategy used by the food industry to amplify their credibility is concealing industry links to information and evidence, including using scientists as advisers, consultants, or spokespersons. In many countries^L(17,18),LM(2),LT(2),LTM(17,26),T(12,21,53,75)^, the food industry used experts to defend private interests without revealing the relationship between them. ILSI was mentioned as one of the spokespersons for the food industry ^L(17,18), LTM(21)^.

Using this created information and biased evidence, as well as due to its supposed credibility and its economic power, the food industry has implemented strategies to influence directly or indirectly the policy discussion. Lobbying was the most cited tactic in the documents, being a barrier for 77.8% in the food taxes cases, 81.2% of the FOPL cases, and 98.7% of the marketing restrictions cases (Appendix A). Additionally, when the food industry failed to avoid policy approval, it acted to delay the discussion^L(8,10,25)^ or negotiated changes to the original, as happened in the Colombia^LTM(7)^ and Ecuador^LT(3)^ FOPL process and in the SSB taxes debate in the Philippines^T(52)^.

Another strategy cited as indirect access was the use of the “revolving door”, that is, when ex-food industry staff work in government organizations and vice versa. The most intriguing cases were of the health minister and health secretary from Colombia and Mexico, respectively, who had a position with food industry companies or organizations funded by them^LTM(21), T(15)^. A president of Mexico had also been president of Coca-Cola in the country and held such a position throughout Latin America^T(38,54)^.

The food industry has also sought involvement in government working, technical or advisory groups, or has provided technical support to policymakers, as described in the Government Environment, Strategies, and Practices topic. Another strategy commonly used was to fund and provide financial incentives to political parties and policymakers to ensure the defense of their interests. The food industry has donated funds to the main political parties during presidential and parliamentary elections^L(17),LTM(7,17,27),M(3),T(55,63)^, and to support government initiatives related to nutrition or health ^LM(2),T(21,55)^. They have also given gifts, funded trips, or donated financial incentives to policy makers^LT(2)^.

One strategy that appeared to be different when comparing policies was the threat to withdraw or withdraw investments from the country if new public policies were introduced. This strategy was predominantly present in food tax discussion. In Denmark, the food industry has affirmed that the taxes would encourage the withdrawal of investments and jobs from the country^T(73)^. In France, Coca-Cola announced that it would discontinue an investment when the SSB tax was approved^T(42,48)^. In Mexico, foreign entrepreneurs and PepsiCo have threatened to move their business to other countries ^T(44,53)^. In addition to these classic threat cases, Coca-Cola decided not to sponsor the Colombian soccer team, as a consequence of the policy^L(17)^.

Concerning legal actions, we have identified examples of litigation to delay the policy process and revoke its approval. In Brazil, the food industry, supported by the advertising sector, was also successful in revoking a marketing regulation in the Court, questioning the legitimacy of the regulatory agency to regulate advertising. Since its suspension, it has not been possible to reverse this^M(5,7),LTM(35)^. In Chile, after the law was approved, the food industry litigated against the State, in defense of their intellectual property, but they had no success^LTM(25)^. The food industry has also requested the suspension of a civil campaign in Colombia, claiming that they were promoting misleading publicity, but as it was evidence-based, the court result was reversed^T(26),LTM(7),L(17)^.

Another strategy related to legal action that the food industry used to create a barrier to these policies’ approval was to influence the development of trade and investment agreements. Despite having been used in all policies, this tactic was implemented in 100% of the FOPL cases, especially because some international forums have an impact on all countries (Appendix A). Documents mentioned the influence of the food industry on Codex Alimentarius ^L(4,11),T(70)^, WTO^L(4,16), LTM(21,23)^, the European Commission and Parliament^L(7,12)^, free trade agreement between Mexico, the USA, and Canada^LT(2)^ and Mercosur^LTM(25)^ negotiations and agreements. According to Thow et al.^L(24)^, more than 60% of Codex observers, who are allowed to contribute to discussions, represent the food industry, which also influences countries’ positions^L(11)^.

### 3.6. Discursive Strategies

The distribution in the discursive strategies category was similar in all policies for both proponents and opponents (Figure 1).

The proponents have focused on the impact of the policy on health, stressing the negative outcomes of obesity and UPF consumption, including its economic costs, and endorsed the policies, promoting benefits to health and to the economy. They have also compared the policies with analogues from other areas, such as tobacco, and the focus on vulnerable populations, particularly infants and children, was an argument used, especially to support food marketing restrictions.

To frame the debate, advocates have presented malnutrition, obesity and NCDs as complex problems. Attention to the causes and solutions for these diseases related to the unhealthy environment, focusing on the industry and its products, highlighting the primacy of profits above health and the unpaid costs by that sector regarding the problems caused. In this regard, restricting advertising, taxing the products, and warning about excessive critical nutrients is seen a way to reduce consumption and consequently improve the health of the population.

Opponents have also focused on the economic and policy impact; they have stressed their own political and financial power, the number of jobs supported, and the money generated for the economy. Thus, contrary to the positive effects presented by the proponents, they argued that the policy approval would reduce their sales and jobs in the country, affecting international business, and that the cost of compliance would be high.

Specifically in food tax cases, the food industry has highlighted the expected costs to society, stressing the possibility of an illegal market or smuggling due to the high cost that the products would have if taxes were approved. The major impact on the poor was emphasized by the food industry. In the USA, the taxes were also framed as a racist policy that had been imposed by white elites ^T(51,74)^.

To frame the debate, the food industry has stressed its good traits, including intentions to support the state with education and collaborate with obesity prevention through non-regulatory measures, corporate social responsibility initiatives, voluntary codes, food reformulation, food security and environmental sustainability. Moreover, unlike civil society, which has focused on the food environment, the food industry has shifted the blame away from its products and has concentrated on individual and parents’ responsibility and the role of sedentary lifestyle in obesity and NCDs. Thereby, the solutions proposed by the food industry were related to the causes of individual problems and were opposed to the paternalist state, i.e., providing information through campaigns or industry-sponsored education, stressing individual control to consume balanced diets, public-private initiatives, especially with the State, and self-regulation with no government participation.

## 4. Discussion

This study demonstrated that the main facilitators of policies aiming to restrict access to UFP are related to the strong role of the State and civil society’s efforts, and the main barriers are a combination of the CPA of the UFP industry and a weak State, vulnerable to commercial interests.

The data are aligned with the Lancet Commission on Obesity report, which concluded that the difficulty in making progress in implementing regulations—related to obesity, malnutrition, and climate change pandemics—is due to inadequate political leadership, combined with strong opposition from powerful commercial interests, and society’s lack of demand for solutions, i.e., political inertia. The balance of power between actors in governance structures determines whether levers of power, such as legal norms, are approved [11]. In this scenario, deregulation and passing on health responsibilities to the individual are central narratives in neoliberal policies and economics, making it difficult to adopt regulations [46]. Therefore, national governments are only able to establish effective governance processes when they have and promote strong commitment from all involved, create coherence between the sectors, ensure sufficient capacity and resources to enable and sustain actions, and implement mechanisms to reduce power imbalances [11].

A mechanism to strengthen governance structures is to invest in capacity, a concept that includes resources, organizational structures, the workforce, partnerships, leadership and governance, knowledge development, and country-specific contexts [47]. Financing and the existence of trained human resources at the governmental level are considered fundamental [11,48]. It is necessary to establish enough professionals trained to deal with the problem and with the necessary skills to conduct collective actions, such as forming coalitions, managing conflicts and processes, carrying out effective communication, and responding to opportunities and threats [11]. Kingdon [45] highlights the role of invisible actors, such as public servants, interest group analysts, parliamentary advisors, and academics, that can affect the governmental agenda in a positive or negative way. They work to publicize the problem and generate and promote alternatives and solutions to decision makers and could be fundamental in each topic being incorporated into the agenda.

This review also identified that progressive scenarios are more favorable to the approval of policies, taxation being an exception in some countries, and thus changes of government in favor of progressive parties can help in the approval of these policies, although this is not a rule. According to Kingdon [45], a problem is considered as part of a political agenda from the conjunction of three flows. One flow is the formation and refinement of policy proposals discussed in a political community. Those which remain in discussion are compatible with the values of its participants, such as their view of the appropriate role or size of government, the appropriate size of the public and private sector, and the importance attributable to correcting society’s inequities, imbalances, or injustices. This last principle of equity is a powerful argument used in debates for or against proposals.

Policies to prevent obesity can have different approaches, such as being based on public health and human rights principles, with greater State intervention to protect the health of the population, or based on the free market, with less State intervention and protection for companies. The last is more lenient, accepted, and easier for governments, because they are embedded in a neoliberal transnational system that promotes government deregulation of markets and reduces social protections, attributing responsibility for health to individuals [11,48]. In this sense, any proposal for intervention that interfere with freedom of business action suffers greater resistance from parties that have this ideology, as seen in this review.

The increasingly concentrated economic power of commercial actors, such as UFP transnationals, is the main source of political inertia, corroborated by this study. Even if the decision to pass laws on policies for healthier and more sustainable food systems remains with the government, political inaction is strengthened by policymakers who believe in educational solutions and economic freedom in dealing with obesity, fear actual or potential opposition from the food industry, avoiding acting against the interests of corporations, are inept and corrupt, or are a combination of these factors [11].

In general, the established structures favor transnational corporations, strengthening power asymmetries through economic liberalism, market deregulation, incentives to expand transnational food and beverage companies, the adoption of self-regulatory actions and the permission (legal or illegal) of the industry’s participation in policy formulation phases, funding for politicians and lobbying public managers [9,11,49,50]. In this scenario, a few companies dominate world sales in the food chain, and these corporations define the pattern of the world’s food system, standardizing eating habits and food culture [9,51,52]. Most of these companies originate from developed countries and seek economic advantages through subsidies, flexible labor legislation and poorly regulated environments, which result in poor and cheap labor and new consumer markets [1,53,54]. Countries with higher levels of urbanization and lower levels of regulation have higher per capita sales of UPF [55], but the volume and variety has been increasing in Asia, Middle East and Africa [9].

Although they can generate jobs and increase income in these countries, as they are transnational companies, they manage to exploit the regulatory differences between them to increase their profits and move from one country to another when they identify economic disadvantages in staying, which can generate, for example, a sudden increase in unemployment rates in some regions [53,56]. To avoid disinvestment, many countries and regions reduce regulation and restrict rights related to social protection, deepening inequalities and reducing their own political role [11,56]. The capacity for financial capital mobility also allows these corporations to threaten to punish governments that initiate regulatory processes that interfere with their profit, which may delay progress or even prevent their adoption by governments [9,52].

The economic power of transnationals, superior to most countries’ economies, makes it possible for them to play an influential role in global governance and to promote their interests in international agreements, influencing the definition of public policies and hindering state action [53,56,57]. Simultaneously, the progressive deregulation of markets and the creation of hybrid governance arrangements, which allow the participation of the food industry in public policies and reduces the accountability of the globalized food industry to governments and national consumers, reinforce the asymmetries of power between these actors [11,46]. These arrangements make up a neo-liberal paradigm that promotes this type of organization and privileges private interests to the detriment of the public interest, by arguing that market deregulation promotes economic growth, and this leads to the reduction of inequalities, which does not occur in practice, but is intrinsic in national institutions and actors [53].

This scenario is a huge challenge to national governments, because the industry’s participation in public policies and its economic importance becomes legitimized and strengthened by these arrangements, threats, and context [11,49]. Although there are discussions about the sector’s contribution to the development of public policies being a legitimate part of democracy, depending on how this occurs, it may actually threaten it [46,49]. There is an imbalance between groups, since there are different organizational capacities and resources in carrying out defense of their interests, which is undesirable in democracies [58]. The lack of resources for civil society to advocate for policies was also identified as a major barrier in this study, for example. Furthermore, while governments must protect public health, the commercial sector has a fiduciary responsibility to maximize shareholder returns [46,48,49]. Higher profits can only be obtained by convincing consumers to buy more or by increasing profit margins in production, for example, by using cheaper ingredients, such as sugar. Thus, higher profits result from UFP, and regulations that create environments that promote healthier choices reduce companies’ revenue. Thus, there is no alignment of interests between public health and the productive sector to enable them to collaborate in the formulation phase of these policies [59].

In countries where there is weak governance and political leadership, corruption, and lack of COI management, food corporations use different CPA strategies to interfere in all stages of public policy management [11,49], as demonstrated clearly in this study. However, lobbying is not the only factor influencing the decision-making process, which also depends on the institutional and political environment, and on the decision-maker’ profile, inclinable or not to lobbying influence [58].

In this sense, the rules, the level of transparency, the economic and sociocultural circumstances, all influence how important lobbying is in defining a public policy [58]. Leadership and political will are imperative for its success and for confronting commercial interests [11,14,48]. It is necessary to create governmental mechanisms that consider health as a priority in relation to commercial results and should include the participation of different sectors that act in a coherent and coordinated manner [14,48]. The linking of efforts from different sectors, led by a government agency with political authority, can broaden commitment, reduce duplication of actions, and make more efficient use of resources. This coordination includes negotiations to generate coherence in the implementation of policies and to reduce power imbalances [11].

To deal with power asymmetries, anti-monopoly laws can be strengthened, increasing competition, to mitigate damages from market concentration; and it is necessary to include independent civil society in the processes of construction and monitoring of policies, providing transparency to the process [46]. In relation to COI, it is essential that they are identified, managed, and minimized, that the actors involved with the industries are transparent and accountable, and that there are no incentives for industries that manufacture products harmful to health [11]. A research carried out to identify the opinions of researchers, activists, and policy makers on appropriate ways to partner with alcohol, tobacco and UPF companies found that most respondents believe that there is a fundamental conflict between private interests and public health objectives, but there is less identification of this conflict when considering food industries. Despite this data, more than 95% of respondents agreed that it is critical for states to adopt policies to increase accountability and transparency in interactions between public sector officials and industries [60].

However, while private sector participation in critical stages of policy development should be avoided, governments still propose public–private partnerships to prevent obesity. These actors can contribute in other stages (e.g., public consultations or audiences), if this is regulated. To support countries in evaluating these partnerships, the WHO [61] and PAHO [62] published an approach and a tool, respectively, for the prevention and management of COI in nutrition policies and programs. These documents consider that policy coherence in the different sectors of government is a key principle of implementation, requiring internal alignment, and that, if partnerships are chosen in any sector, there must be a guarantee that they do not jeopardize the nutritional goals.

Another challenge for the governments and private sector is to create new business models that help to reduce negative externalities created by the current economy. This would be an ethical and democratic way for the private sector to contribute to the food system and its regulation. The social responsibility practices used by corporations are a form of marketing, not implying real responsibility on the part of the company. If performance targets and criteria were transparent, independently monitored and there was a financial impact on companies depending on the increase or decrease in externalities, concrete changes could occur in food systems [11]. Lobbying regulation can also be a way for groups to act positively for democracy, contributing to transparency in the relationship processes between groups and decision makers and avoiding capture by private interests. In these regulations, it is essential that there are principles and rules of conduct for interacting with lobbyists, prohibiting the use of power by decision-makers to obtain advantages, receiving gifts/favors and invitations to social events, disclosing amounts received, or providing services, such as consulting and advisory services. Transparency in investment, income, shareholding in private companies, links with organizations, publication of agendas and reports of meetings, which should be accompanied by public servants, and the guarantee of the right to opposition are also criteria to be included in these regulations [46,63].

Furthermore, another challenge to governments is to establish accountability mechanisms, which are an essential part of policy implementation, even when there is no private participation [11,46,48]. Well-planned, executed and strengthened accountability can ensure better policy progress, better performance by the private sector and restriction of its influence, and greater transparency of actions, strengthening civil involvement. For this, there must be an independent assessment by uninvolved and reliable institutions, with authority and execution capacity, since accountability requires that one actor responds to another and that there are sanctions in case of non-compliance [11,46].

In that sense, regulations also need to be constantly evaluated regarding their impact, and this step must be included in the formulation of policies [12]. In this sense, participation and partnerships with the academic sector are essential, since the evidence arising from the evaluation of implemented policies also plays an important role in obtaining support from society and policy makers [36].

The basis of scientific evidence without COI was also a facilitator found in several studies. When a country requests support from academic groups to evaluate the best proposals, avoiding the participation of industry in the process, there is a greater chance of success in approval, as occurred in Chile and Mexico ^LT(3),LTM(8)^. In this sense, another challenge for civil and academic society is to maintain the independence of their research, when there is little or no public funding, considering that they are already underfunded, as demonstrated in this review. On the other hand, governments must assume their role in research funding to compete with private finance and guarantee impartiality in the country’s scientific production. In addition, even when there is funding, they face the challenge of ensuring that the research used, and the researchers invited to policy-making spaces, do not have conflicts of interest with the food industry.

This issue is fundamental, as there are institutions financed by the food industry that promote results favorable to their interests, as in the already mentioned case of ILSI. Despite being funded by several transnational food companies, ILSI uses the narrative of impartiality. However, some studies describe the work of this institute as redirecting the formulation of regulations using evidence produced by them—and financed by companies, such as Coca-Cola—and prepared for policy makers. ILSI emerged in 1978 and has been expanding as a response to criticism of research funding by the food industry and has managed to influence the scientific production and principles of scientific integrity of the food science and nutrition community, building trust in this field. As a result, it promotes and participates in scientific events, and has been invited to compose policy formulation groups [64,65], including those related to regulation, as mentioned in this review.

This type of biased evidence, which is mainly based on nutritionism, only benefits the food industry, because it shifts discussions from the context of the food system, food culture, socioeconomic contexts, the responsibilities of manufacturers and the government itself, to focus on isolated and fragmented foods and nutrients. This diverts attention from all the factors and ingredients that involve the manufacture of UFP, as well as the damage caused by these products to health, the environment, culture, and, consequently, confuses discussions about the need to regulate these products and of the practices of the industries involved in its chain [66]. Thus, institutions such as ILSI and funded research by the food industry should not be treated by governments, civil society, and international organizations as an impartial scientific community, but as cautiously as if they were the industry itself.

Evidence-based arguments by civil society are critical and an important counterpoint to industry attempts to block policy approval. The evidence enables greater capacity to defend the chosen policies, facilitating negotiations and providing credibility to frame the public debate. Thus, partnerships between academics, policymakers and other advocates contribute to the transformation of theoretical implementation estimates into policies, map barriers, and direct researchers to choose relevant objects for policy definition and implementation [11,36].

Civil society was identified as the second most important component in promoting the approval of regulations, and is strengthened by an organized, coordinated, and cohesive approach. However, the studies presented that, while public health and consumer rights professionals generally support the issue, their organizations are often characterized as small, fragmented, uncoordinated and lacking in support funding to generate sufficient demand to pressure governments and overcome opposition from industry, which has disproportionate economic and political power [11,49]. The defense of interests depends on the existence of resources [58], which is a considerable challenge for civil society. When substantial funding is present, as in the case of Mexico, it increases the chance of adopting different strategies for a more equitable discussion and approval of the policy. This data reinforces the importance of governments creating mechanisms for participation, as well as investment in and strengthening of civil society capacities, guaranteeing their autonomy, in order to improve the balance of power in the formulation of public nutrition policies.

This means not only creating formal mechanisms for social participation, but also guaranteeing effective popular participation in the formation of the studied policies, as well as allowing influence on the main state decisions related to the right to health and to food, in all spheres of state power (Executive, Legislature and Judiciary). Real democratic participation requires the establishment of legally regulated rules and procedures for organizing its functioning [67].

In this sense, it is important that the decisions taken by public agents, in the exercise of their specific state functions, must be anchored in the broad participation of the people in the democratic process, in order to allow the permeability of all state decisions (administrative, legislative and judicial) to the will of the people. In this way, health democracy requires the construction of institutions that allow effective popular participation at all levels and spheres of state power, which involves the elaboration of laws that regulate matters related to health, the creation of public policies with the objective of giving effect to the right to health and food and, also imply in judicial decisions to repair concrete damage to health perpetrated by public or private agents [67]. 

Joining forces with other movements, such as those focused on consumers’ rights or climate change, is a strategy that can create pressure towards the adoption of more effective actions [11,48], as seen in several examples presented in this study. Not only should the policies address the syndemic of obesity, undernutrition, and climate change, but social movements should also act together to strengthen the pressure [11]. Considering that the activities of food companies are transnational, another strategy is learning from other policies, such as tobacco policy, cooperating with movements from other countries on the same policy and obtaining allies from academia and international organizations to exert pressure. This is a portrait of the social movements of the 21st century, which are becoming international, with broader targets, in the form of transnational corporations, creating ties with international organizations, and relationships with movements around a specific cause and acting in a coordinated way [68], to counteract the power of these companies.

Finally, despite being less mentioned, international organizations have played an important role in inducing regulatory policies in some countries, mainly in the Americas. It is not possible to identify whether the few mentions were due to the lower number of reported cases or lower performance in other countries. However, organizations such as PAHO, WHO, and the WB were important in providing technical and even political support to some countries, as well as in inducing the agreement of regional and global commitments.

The WHO, for example, has been considered as scientific and technical rather than an authority on global health. One reason may be the need not to conflict with international private donors, considering that around 60% of the budget comes from voluntary contributions (from the countries themselves, philanthropic foundations, and private companies). However, these organizations can still be leveraged by countries to promote solutions in line with their national identity and interests, based on their recommendations for nutrition policies. In addition, spaces for exchanging experiences on policies and agreeing on commitments are ways of creating alliances, especially regional ones, to overcome health problems and determinants that transcend national borders, such as the actions of transnational corporations [69]. As a result, it is possible to influence the global agenda and develop political and technical instruments for global and regional governance favorable to the approval of the measures discussed in this study, such as by the establishment of a framework convention, such as the existing one on tobacco [11].

This is a very important role, since international agreements related to the subject studied are voluntary and there are almost no supervisory bodies and inspections of their compliance with international trade and investment agreements which bind transnational companies. National rules are insufficient to deal with these treaties, as countries are poorly protected in relation to the activities of these corporations, unlike foreign investment, which is protected by treaties signed between countries and companies, protecting the investor [46,56,70]. In addition, while civil society has limited ability to participate in discussion forums on these agreements, interested transnationals, such as the food and pharmaceutical industries, combine their powers and often work with delegations from their own countries [70], such as the mentioned case from the Codex. These discussions reinforce the need to strengthen democracies and mechanisms to promote real social participation in food policy definition.

In that sense, although a review is focused on the three policies, it is recognized that, even if they are important, they only deal with actions limited to adjusting or reforming distortions in national food systems (cost of food, consumer information, abusive advertising) instead of transforming the global food system. For there to be a real reduction in UFP consumption, with positive impacts for people and the planet, a broader, intersectoral, coordinated, multilevel, democratic, and participatory approach is also essential, which makes it possible to act on the various determinants of the UFP [71]. Thus, it is the toleration of extractive, materialist and neoliberal governance systems, structures, practices and beliefs that underpin capitalism and prioritize unbridled economic growth, generating increased consumption, including products that cause harm to health [11]. This is only possible with the establishment of democratic and participatory democracies that promote the inclusion of all stakeholders but ensure balance and redistribution of power in political discussions and throughout the food system [71].

This study has some limitations. It was not a detailed analysis of all factors, because it was limited to articles and documents available relating to its references. Even with a broad search and reference review, some documents may not have been found. In addition, the fact that some cases have been more commonly studied than others is an important limitation, as well as the lack of embracing studies that consider all policy process aspects. This could have biased the results, since there were many papers included that have focused on food industry practices. Moreover, a factor that has not been identified does not mean that it has not occurred, but it may not have been described by the selected articles, which can only be resolved with a large multi-center study that includes interviews in all countries. A statistical analysis would also be interesting to identify the contribution of each factor to the approval or non-approval of the policies.

Despite these limitations, this is an important study that consolidates and compares data about barriers and facilitators related to regulatory and fiscal nutrition policies discussion and approval. It can elucidate ideas for the actors from the countries where these three policies are being discussed and provide possible ways and strategies to open and to facilitate the identification of windows of opportunity in different realities and contexts.

## 5. Conclusions

This scoping review study consolidated barriers and facilitators related to the adoption of policies to reduce ultra-processed foods consumption, demonstrating that factors related to the actions of governments and civil society are the main facilitators of the discussion and approval of FOPL, food marketing restrictions and taxation policies. On the other hand, being the most interested actor in promoting the consumption of these products, the strategies adopted by the companies that produce UFP constitute the main barrier to these policies in all the studied countries.

Although the country’s political and economic scenario, as well as factors related to knowledge and scientific production on the subject are fundamental, governments have an essential role in recognizing the problem, and must lead the policy discussion and the search for existing alternatives, including funding research to promote evidence construction without COI, to counter-argument false and biased discourses constructed by policy opponents. As enablers, governments must have sufficient leadership to avoid the interference of commercial interests in the definition of public policies, not only related to the subject studied, but to all health policies.

In this sense, our review found that the main barrier, that is, political inertia resulting from CPA strategies carried out by UFP companies, must be overcome to approve the studied policies. It is necessary to create mechanisms to mitigate and prevent COI, regulate lobbying and reduce food company’s political power. At the same time, it is fundamental to promote the participation of independent civil society in policy discussion, as well as its strengthening and its survival, by financial and legal structures that ensure their influence and participation in policy decisions. In this regard, international institutions and organizations could also contribute, funding civil society, creating barriers to the participation of the private sector in the definition of international recommendations and agreements and promoting discussions that lead to the creation of international norms such as the Tobacco Framework Convention, to induce the approval of policies by countries that have encountered difficulties. An ecological approach to regulation of the food system is also emerging and is critical to policy thinking and direct action on broader determinants of global health and nutrition.

## Figures and Tables

**Figure 1 ijerph-20-04729-f001:**
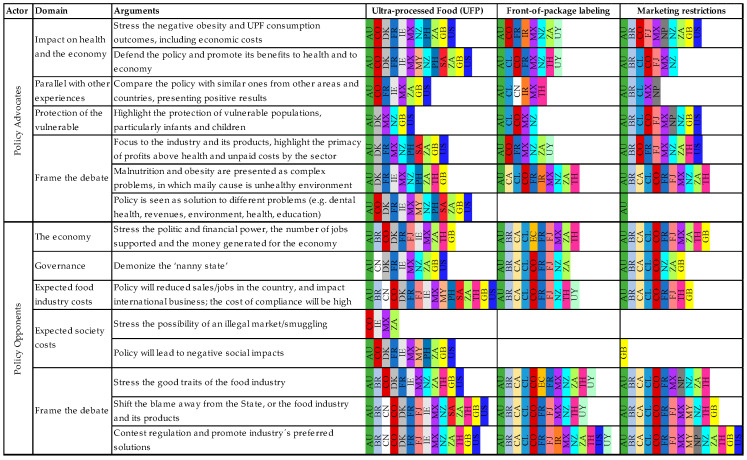
Discursive strategies by actors and country.

**Figure 2 ijerph-20-04729-f002:**
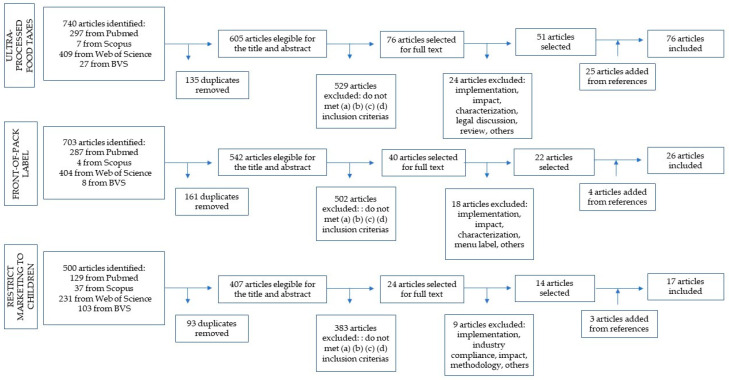
Scoping review flowchart.

**Table 1 ijerph-20-04729-t001:** Search terms used by studied policy and database. 2022.

Policy	Search Terms
Ultra-processed food and beverage tax	PUBMED: (((tax OR price)) AND (“unhealthy food” OR soda OR “sugar sweetened beverage”)) AND (barrier OR enabler OR difficult OR facilitator OR “food industry” OR “big food” OR “political choice” OR reinforcement OR implement OR experience))
	Web of Science (WOS): TS = ((tax OR price) AND (unhealthy food OR soda OR sugar sweetened beverage) AND (barrier OR enabler OR difficult OR facilitator OR food industry OR big food OR political choice OR reinforcement OR implement OR experience))
	Scopus: ((tax OR price) AND (unhealthy AND food OR soda OR sugar AND sweetened AND beverage) AND (barrier OR enabler OR difficult OR facilitator OR food AND industry OR big AND food OR political AND choice OR reinforcement OR implement OR experience))
	Virtual Health Library (BVS): impostos OR preço OR tributo) AND (“alimentos não saudáveis” OR refrigerantes OR “bebidas adoçadas” OR “bebidas açucaradas”) AND (“indústria de alimentos” OR “indústria alimentícia” OR implementação OR experiência)
Front-of-package labeling	PUBMED: ((“front-of-package” OR “traffic light” OR “warning label” OR GDA OR “Nutri-score” OR CNL OR “star rating”) AND (“food label” OR “food labeling”) AND (barrier OR enabler OR difficult OR facilitator OR “food industry” OR “big food” OR “political choice” OR reinforcement OR implement OR experience)
	WOS: TS = ((front-of-package OR traffic light OR warning label OR GDA OR Nutri-score OR CNL OR star rating) AND (food) AND (label OR labeling) AND (barrier OR enabler OR difficult OR facilitator OR food industry OR big food OR political choice OR reinforcement OR implement OR experience))
	Scopus: ((front-of-package OR traffic light OR warning label OR GDA OR Nutri-score OR CNL OR star rating) AND (food) AND (label OR labeling) AND (barrier OR enabler OR difficult OR facilitator OR food industry OR big food OR political choice OR reinforcement OR implement OR experience))
	BVS: (rotulagem nutricional frontal OR semáforo OR alerta OR advertência OR GDA OR Nutri-score OR CNL OR star rating) AND rotulagem AND alimentos AND (“indústria de alimentos” OR “indústria alimentícia” OR implementação OR experiência)
Marketing restriction	PUBMED: (((marketing OR promotion OR placement OR advertising OR television OR media) AND (children OR child) AND (“unhealthy food” OR “ultra-processed”)) AND (regulation OR policy) AND (barrier OR enabler OR difficult OR facilitator OR “food industry” OR “big food” OR “political choice” OR reinforcement OR implement OR experience))
	WOS: TS = ((marketing OR promotion OR placement OR advertising OR television OR media) AND (unhealthy food OR ultra-processed) AND (regulation OR policy) AND (barrier OR enabler OR difficult OR facilitator OR food industry OR big food OR political choice OR reinforcement OR implement OR experience) AND (children OR child))
	Scopus: TITLE-ABS-KEY (((marketing OR promotion OR placement OR advertising OR television OR media) AND (unhealthy AND food OR ultra-processed) AND (regulation OR regimentation OR policy) AND (barrier OR enabler OR difficult OR facilitator OR food AND industry OR big AND food OR political AND choice OR reinforcement OR implement OR experience) AND (children OR child)))
	BVS: (marketing OR publicidade) AND (alimentos OR ultraprocessados) AND (crianças OR infância OR infantil)

**Table 2 ijerph-20-04729-t002:** Number and percentage of documents characteristics by food policy.

Policy	Food Tax i		Front-of-Package labeling i		Marketing Restrictions i		Total	
Characteristics	N	%	N	%	N	%	N	%
Publication Year								
2005–2009	3	2.4	2	2.8	4	6.5	4	2.4
2010–2013	22	17.6	15	20.8	16	25.8	32	19.0
2014–2017	41	32.8	15	20.8	15	24.2	48	28.6
2018–2022	59	47.2	40	55.6	27	43.5	84	50.0
Document								
Article	100	80.0	58	80.6	57	93.4	131	78.0
Report	7	5.6	2	2.8	2	3.3	7	4.2
Review	5	4.0	3	4.2	0	0.0	6	3.6
Scientific News	5	4.0	0	0.0	1	1.6	6	3.6
Others	8	6.4	9	12.5	1	1.6	18	10.7
Experience/Support document								
Experience	91	71.7	56	80.0	45	71.4	130	77.4
Support	36	21.3	14	20.0	18	28.6	38	22.6
Period covered ^i^								
1–5 years	91	74.0	45	62.5	40	63.5	112	66.7
5–10 years	10	8.1	12	16.7	9	14.3	21	12.5
More than 10 years	10	8.1	10	13.9	12	19.0	18	10.7
Not identified	12	9.8	5	6.9	2	3.2	17	10.1
Subject								
Process policy	44	64.7	22	71.0	22	75.9	64	38.1
Strategies used by stakeholders	32	47.1	28	90.3	18	62.1	46	27.4
Population support	33	48.5	14	45.2	17	58.6	37	22.0
Stakeholders’ opinion	7	10.3	3	9.7	3	10.3	7	4.2
Media analysis	7	10.3	0	0.0	2	6.9	9	5.4
Codex or World Trade Organization discussions	1	1.5	3	9.7	1	3.4	3	1.8
Role of scientific evidence	1	1.5	2	6.5	0	0.0	2	1.2
Coverage								
Global	2	1.6	3	4.1	1	1.5	4	2.4
Regional	0	0.0	5	6.8	5	7.6	9	5.4
National	101	82.8	66	89.2	55	83.3	136	81.0
Local	19	15.6	0	0.0	5	7.6	23	13.7
Country ^i,iv^								
Australia	22	14.6	22	21.4	19	32.2	32	14.5
Brazil	4	2.6	5	4.9	5	8.5	8	3.6
Canada	2	1.3	3	2.9	2	3.4	4	1.8
Chile	7	4.6	9	8.7	6	10.2	8	3.6
China	0	0.0	2	1.9	0	0.0	2	0.9
Colombia	4	2.6	5	4.9	3	5.1	6	2.7
Denmark	7	4.6	0	0.0	0	0.0	7	3.2
Ecuador	0	0.0	3	2.9	0	0.0	3	1.4
France	9	6.0	14	13.6	0	0.0	17	7.7
Fiji	2	1.3	0	0.0	3	5.1	4	1.8
Iran	0	0.0	2	1.9	0	0.0	2	0.9
Ireland	6	4.0	0	0.0	0	0.0	6	2.7
Malaysia	2	1.3	0	0.0	1	1.7	3	1.4
Mexico	16	10.6	5	4.9	2	3.4	20	9.0
Nepal	0	0.0	0	0.0	1	1.7	1	0.5
New Zealand	6	4.0	8	7.8	0	0.0	10	4.5
Peru	0	0.0	3	2.9	0	0.0	3	1.4
Philippines	3	2.0	0	0.0	0	0.0	3	1.4
Saudi Arabia’s	1	0.7	0	0.0	0	0.0	1	0.5
South Africa	9	6.0	4	3.9	3	5.1	11	5.0
Thailand	3	2.0	0	0.0	0	0.0	3	1.4
United Kingdon ^ii^	16	10.6	8	7.8	4	6.8	24	10.9
USA ^iii^	32	21.2	6	5.8	10	16.9	39	17.6
Uruguay	0	0.0	4	3.9	0	0.0	4	1.8

^i^ Some documents present data from more than one country, or more than one policy. Therefore, the number of documents can be duplicated; ^ii^ Marketing restriction data from the UK and London; ^iii^ Tax data from the Federal Government and 15 jurisdictions (Albany, Berkeley, Boulder-CO, Cook, El Monte, Kansas, New York, Oakland, Philadelphia-PEN, Richmond, Seattle-WA, Santa Fe, San Francisco, Telluride); ^iv^ The total percentages were calculated based on the sum of the number of times that the country appeared.

**Table 3 ijerph-20-04729-t003:** Number and percentage of document characteristics by food policy.

Category	Domain	Food Tax				Front-of-Package Labeling				Marketing Restrictions				All Policies			
		Enablers		Barriers		Enablers		Barriers		Enablers		Barriers		Enablers		Barriers	
		N	%	N	%	N	%	N	%	N	%	N	%	N	%	N	%
Contextual	Political-economic environment	27	9.4	31	9.3	13	6.4	19	6.9	13	6.5	18	6.3	53	7.7	68	7.6
	Knowledge and awareness	18	6.3	3	0.9	15	7.4	4	1.4	14	7.0	6	2.1	47	6.8	13	1.5
	Total	45	15.7	34	10.2	28	13.8	23	8.3	27	13.5	24	8.4	100	14.5	81	9.1
Government	Government environment	70	24.5	35	10.5	38	18.7	19	6.9	33	16.5	17	6.0	141	20.5	71	7.9
	Governance	56	19.6	30	9.0	52	25.6	28	10.1	46	23.0	32	11.2	154	22.4	90	10.1
	Coalition management	16	5.6	12	3.6	14	6.9	9	3.2	12	6.0	13	4.6	42	6.1	34	3.8
	Workforce and funding	2	0.7	4	1.2	2	1.0	3	1.1	2	1.0	7	2.5	6	0.9	14	1.6
	Technical issues	0	0.0	3	0.9	0	0.0	3	1.1	0	0.0	4	1.4	0	0.0	10	1.1
	Total	144	50.3	84	25.2	106	52.2	62	22.4	93	46.5	73	25.6	343	49.8	219	24.5
International Organizations	Political environment	9	3.1	0	0.0	8	3.9	0	0.0	6	3.0	0	0.0	23	3.3	0	0.0
	Involvement and influence in policy	3	1.0	0	0.0	4	2.0	0	0.0	3	1.5	0	0.0	10	1.5	0	0.0
	Total	12	4.2	0	0.0	12	5.9	0	0.0	9	4.5	0	0.0	33	4.8	0	0.0
Civil Society	Information management	26	9.1	0	0.0	16	7.9	0	0.0	17	8.5	0	0.0	59	8.6	0	0.0
	Involvement and influence in policy	18	6.3	0	0.0	19	9.4	0	0.0	20	10.0	0	0.0	57	8.3	0	0.0
	Coalition management	38	13.3	9	2.7	21	10.3	4	1.4	30	15.0	5	1.8	89	12.9	18	2.0
	Legal actions	0	0.0	0	0.0	0	0.0	0	0.0	3	1.5	0	0.0	3	0.4	0	0.0
	Resources	0	0.0	2	0.6	0	0.0	2	0.7	0	0.0	3	1.1	0	0.0	7	0.8
	Total	82	28.7	11	3.3	56	27.6	6	2.2	70	35.0	8	2.8	208	30.2	25	2.8
Private Sector	Information management	0	0.0	59	17.7	0	0.0	52	18.8	0	0.0	54	18.9	0	0.0	165	18.4
	Involvement and influence in policy	0	0.0	46	13.8	0	0.0	39	14.1	0	0.0	45	15.8	0	0.0	130	14.5
	Legal actions	0	0.0	16	4.8	0	0.0	22	7.9	0	0.0	10	3.5	0	0.0	48	5.4
	Coalition management	3	1.0	83	24.9	1	0.5	73	26.4	1	0.5	71	24.9	5	0.7	227	25.4
	Total	3	1.0	204	61.3	1	0.5	186	67.1	1	0.5	180	63.2	5	0.7	570	63.7
**Total**		**286**	**46.2**	**333**	**53.8**	**203**	**42.3**	**277**	**57.7**	**200**	**41.2**	**285**	**58.8**	**689**	**43.5**	**895**	**56.5**

## Data Availability

All data can be accessed by following the methodology described in this paper and in the documents included in this review. To detailed information, contact the corresponding author.

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
