# Peer review of "Barriers and Facilitators Related to the Adoption of Policies to Reduce Ultra-Processed Foods Consumption: A Scoping Review"

_ijerph, 2023, doi:10.3390/ijerph20064729_

Round 1
Reviewer 1 Report
some typos, and grammer please check
please mention what ILSI has been done, with detail information
discussion should be enriched, how about challenges for government, academia, and industry partnerships ? possible harmonization for regulatory on ultra processed foods, and more references should be for a review paper
conclusion does not match the title
Reviewer 2 Report
Dear authors
thank you for presenting a very interesting and actuall topic
The general opinion of the paper is:
it is well written, with proper introduction, methods , discusion and conclusions according to the topic od research.
The possible minor changes could be made in abstract:
describe conclusions that go pro or contra (barriers /facilators) to policies , that you have already stated in conclusion section4.
such as effective communication to public, policies for preventing obesity, country specifc policies ,,.... etc
AND In section 3.4. you can divide pro et contra strategies,
as you mentioned that Usa regulations are not so strict, pressures of food industry, influence on policy
v.s.
European strategies (France ....) that are more strict and concerned about health issues and food
with higher public awareness about health and food
best regards,
the reviewer
